# High-Fat Foods and FODMAPs Containing Gluten Foods Primarily Contribute to Symptoms of Irritable Bowel Syndrome in Korean Adults

**DOI:** 10.3390/nu13041308

**Published:** 2021-04-15

**Authors:** Woori Na, Yeji Lee, Hyeji Kim, Yong Sung Kim, Cheongmin Sohn

**Affiliations:** 1Department of Food and Nutrition, Wonkwang University, Iksan, Jeonbuk 54538, Korea; nawoori6@gmail.com (W.N.); call960823@naver.com (Y.L.); xzc0206@naver.com (H.K.); 2Institute of Life Science and Natural Resources, Wonkwang University, Iksan, Jeonbuk 54538, Korea; 3Gastroenterology and Digestive Disease Research Institute, School of Medicine, Wonkwang University, Iksan, Jeonbuk 54538, Korea; wms89@hanmail.net

**Keywords:** irritable bowel syndrome, dietary assessment, FODMAP, high fat, glutens, Korea

## Abstract

Dietary control plays an important role in the treatment of irritable bowel syndrome (IBS). However, few studies have examined the relationship between dietary intake and symptoms of IBS in Koreans. The current cross-sectional study aimed to examine the diet in food consumption and nutrient intake in Korean adults aged 20 to 40 with IBS. The data collected were completed by 857 subjects using a community-based web survey. The questionnaire covered functional bowel disorders based on Rome III, the semi-quantitative Food Frequency Questionnaire (SQ-FFQ), and the food items causing symptoms. In total, 186 of 857 subjects (21.7%) were diagnosed with IBS. The non-IBS group had a fat intake of 76.9 ± 47.9 g/day, while the IBS group had a fat intake of 86.6 ± 55.1 g/day (*p* = 0.014). The non-IBS group had a total fermentable oligosaccharide, disaccharide, monosaccharide, and polyol (FODMAP) intake of 12.6 ± 9.7 g/day, whereas the IBS group had a total FODMAP intake of 13.9 ± 9.9 g/day (*p* = 0.030). Foods that contributed to the onset of symptoms in the IBS group were instant noodles (70.8%), Chinese noodles with vegetables and seafood (68.7%), pizza (67.2%), and black bean sauce noodles (66.3%) which are mostly classified as high fat and high gluten foods. The dietary intake of IBS patients differs from that of non-IBS subjects. Increased intake of gluten-containing or high-fat foods due to the westernized diet caused more IBS symptoms than high FODMAPs and dairy products in Korean adults in their 20 s to 40 s.

## 1. Introduction

Irritable bowel syndrome (IBS) is a chronic disease characterized by repeated symptoms such as abdominal discomfort, abdominal pain, diarrhea, and constipation, without an underlying disease that affects the gastrointestinal area [1]. Such symptoms reduce the quality of life as they interfere with day-to-day living and result in reduced work productivity, activity intolerance, and increased medical expenses [2,3]. IBS has a global prevalence rate of 9–20%, and of 6.6–26.7% in Korea [4,5,6,7]. However, the pathogenesis of IBS is not clearly understood, and its treatment is aimed at alleviating symptoms. The factors that influence IBS symptoms include changes in intestinal motility/sensitivity, impaired control of the brain–gut axis, psychosocial/genetic factors, changes in the intestinal flora, stress, and dietary factors [8,9,10,11]. Based on previous studies, the British Dietetic Association reported that alcohol, caffeine, spicy food, fat, water in food, dietary habits, dairy products, dietary fiber, gluten, and high-sensitivity food may aggravate the symptoms [12]. A diet with low levels of fermentable oligosaccharide, disaccharide, monosaccharide, and polyol (FODMAP) has been recommended for IBS patients as the highest-level evidence-based dietary guideline. Since short carbohydrates are not easily absorbed within the intestine, they increase the osmotic pressure or are easily broken down by enterobacteria, causing an increase in intestinal gas [13,14]. Studies on food ingredients and irritable bowel syndrome are continuously actively conducted [15]. In Korea, although the prevalence rate of IBS exceeds the global average and the occurrence of IBS is closely related to individuals’ dietary intake, not many studies on its relationship with food intake have yet been conducted, and baseline data for nutritional treatment are still insufficient. Korean meals consist of staple foods and side dishes made with a variety of ingredients and seasonings. Therefore, the types of food consumed might be relatively diverse compared with those consumed in Western countries. Koreans consume large amounts of vegetables in kimchi or side dishes, which infers the consumption of high FODMAPs. In this study, the dietary status of Korean adults in their 20 s to 40 s with IBS was examined, and their intake of FODMAPs and other types of food components that contribute to IBS symptoms was analyzed, to provide baseline data for establishing dietary guidelines for Korean patients with IBS.

## 2. Materials and Methods

### 2.1. Study Participants

A community-based random web survey of the general population in their 20 s to 40 s was conducted from April 2020 to August 2020. We excluded subjects with inflammatory bowel disease and those who take drugs that can severely affect gastrointestinal motility (antibiotics, enema, anticonvulsant, anti-psychotic, and Parkinson’s treatment). This study was approved by the Bioethics Committee of Wonkwang University (WKIRB-202004-SB-012). All participants provided electronic informed consent. One thousand individuals (male: 497 participants, female: 503 participants) completed the survey. We used data acquired from 918 participants in the study after excluding participants who responded insincerely, with missing data in the questionnaire such as IBS diagnosis (*n* = 10), dietary intake data, or symptoms of ingested food (*n* = 57), stress/depression/anxiety (*n* = 15). We also excluded the data obtained from 41 participants whose dietary intake was less than 500 kcal and data from 20 participants whose dietary intake exceeded 9000 kcal; consequently, finally, 857 participants were used for analysis.

The survey consisted of five questions relating to IBS classification, food items, and menus consumed when the IBS symptoms developed; 118 food items and menus relating to food intake frequency; 5 questions relating to stress; 14 questions relating to anxiety/depression; 8 questions relating to physical activities; and 11 questions relating to general characteristics. The Korean version of the Rome III criteria questionnaires developed by Drossman et al. was used to classify IBS [16,17]; participants who had been showing symptoms for over 6 months and had experienced recurrent abdominal pain 1 day per week for the last 3 months, which is associated with at least 2 of the 3 criteria related to defecation, were classified with IBS.

### 2.2. Dietary Data

Participants completed a web-based semi-quantitative food frequency questionnaire (SQ-FFQ) about their dietary intake over the past six months. The questionnaire was a revised form of SQ-FFQ of Korea National Health and Nutrition Survey (KNHENES) questionnaire, validated for energy and nutrient analysis [18,19]. Among the 112 items in the KNHENES’s FFQ, the item with low FODMAPs and low intake levels was excluded (boiled lotus root; 1 item). Two types of grilled fish with similar inclinations were queried with one item (grilled mackerel∙ grilled yellow croaker) and among Korean type Chinese foods, those with different cooking methods were classified into two items (Chinese-style noodles vegetables and sea-foods and black bean sauce noodles). Additionally, three food items with a high content of FODMAPs were added (mango, dried fruits, canned fruits). In reflection of the Korea nutritional statistics analysis and food industry statistics, four food items mainly consumed by adults in their 20 s to 40 s were added [20] (French fries, bowl of rice, spaghetti, meat ball). Finally, the questionnaire was composed of 118 items. The symptom-related questionnaire was prepared through consultation and review by six gastroenterologists. In order to determine the symptoms according to the intake of food items, each of the 118 items was asked whether there was any experience with “no symptoms”; “bloating”; “diarrhea”; “constipation”; and “pain”, which are the same symptoms as those of the Rome III criteria. The questionnaire regarding symptom patterns was not validated for IBS patients. When one of the food items caused the occurrence of symptoms, the case was considered “symptom shown”. According to the British Dietetics Association (BDA) guidelines, we reclassified 118 foods into seven food types that caused IBS symptoms [12]. The seven food types are as follows; high FODMAP content (37 food items/menus), high fat content (30 food items/menus), high FODMAPs containing gluten content (19 food items/menus), dairy products (5 food items/menus), high caffeine beverage (5 food items/menus), alcohol (3 food items/menus), and spicy (10 food items/menus) (Appendix A). We defined high FODMAP content based on Monash University’s internet-based program for FODMAP diet for IBS [21]. In addition, high FODMAP contents were determined as considering each food’s serving size of high FODMAPs presented by Monash University’s internet-based program for FODMAP diet for IBS. High-fat foods were considered to have a fat content of 30% or more of total energy in food (based on Dietary Reference Intake for Korean (KDRI)). High FODMAPs containing gluten were defined as foods mainly made of flour such as noodles, bread, wheat cakes, dumplings, and cookies based on research by Lee et al. [22]. Regarding the nutrient composition, the 9.2 revision of the National Standard Food Composition data published by the Rural Development Administration in Korea was utilized in this study [23]. Furthermore, the FODMAPs database was complemented by adding seven foods that are high in fructose, 43 foods of sorbitol, 39 foods of mannitol, 11 foods of raffinose, 12 foods of stachyose, and 42 foods of fructan based on the previously published papers [24,25,26,27,28]. Total FODMAPs was calculated by adding the content of fructan, galacto-oligosaccharides (GOS), lactose, excess fructose, solbitol, and mannitol.

### 2.3. Health Status and General Characteristics

The Korean version of Brief Encounter Psychosocial Instrument (BEPSI-K) revised by Kim et al. [29] was used to evaluate the negative effects of stress on health, and the items were rated using a 5-point scale, ranging from 1 as “not at all” to 5 as “very frequently.” The points for each item were added, and the total points were divided by 5 to calculate the mean value. Scores below 1.8 points were classified as low stress, scores ranging from 2.0 to 2.6 points were classified as middle stress, and scores above 2.8 points were classified as high stress. The Hospital Anxiety and Depression (HAD) revised by Oh Se Man et al. [30] was used to measure the symptoms of anxiety and depression. It contains 14 questions (7 questions rated using the HAD-A (Anxiety) scale and 7 questions rated using the HAD-D (Depression) scale) and consists of two subscales: anxiety/depression. Each item is rated using a 4-point scale (0–3 points). The scores ranging from 0 to 7 points indicate normal, scores ranging from 8 to 10 points indicate mild anxiety/depression, scores ranging from 11 to 14 points indicate moderate anxiety/depression, and scores ranging from 15 to 21 points indicate severe anxiety/depression. The Korean version of the International Physical Activity Questionnaire (IPAQ), developed by Oh et al., was used to evaluate the level of physical activity [31]. The walking metabolic equivalent task (MET) value (min/week) was calculated using the following formula: 3.3 (MET level) × walking time (min) × day (day). The moderate activity MET value was calculated using the formula 4.0 (MET level) × moderate activity time (min) × day (day), while the intense activity MET value was calculated using the formula 8.0 (MET level) × intense activity time (min) × day (day). The calculated points were summed up; scores below 600 MET were classified as no activities, scores ranging from 600 to 3000 MET were classified as minimum activities, and scores above 3000 MET were classified as health-enhancing physical activities. Data on the following general characteristics were obtained: age, sex, height, weight, occupation, place of residence, drinking status, smoking status, dietary supplement use, food allergies and medical history of related gastrointestinal diseases such as gastrointestinal reflux disease, functional dysphasia, chronic constipation, functional diarrhea, stomach cancer (cured), and colon cancer (cured); the patient’s height and weight were used to calculate the body mass index (BMI) (in kg/m^2^).

### 2.4. Statistical Analysis

Based on the general characteristics and health status of the participants classified with IBS, a crossover analysis was performed to analyze the categorical variables such as sex, obesity ratio, current drinking, current smoking, use of dietary supplement, medical history, diagnosis of food allergy, stress, anxiety, and depression. Additionally, the Mann–Whitney test was used to compare continuous variables, such as age, body mass index, including nutrient intake. In addition, we used a frequency analysis to determine the contribution rate of food/menu and food group related to symptoms and expressed it as n (%). Statistical analysis was performed using SPSS (v26.0, Statistics Package for Social Science, IBM Corp., Armonk, NY, USA). All data were expressed as the mean and standard deviation, and *p* values < 0.05 indicated significant differences.

## 3. Results

### 3.1. General Characteristics of the Study Participants

The general characteristics of the study participants are listed in Table 1. The control group had 350 male participants (52.2%); meanwhile, the IBS group had 78 male participants (41.9%), which was significantly lower than the number of male participants in the control group (*p* = 0.016). The control group had 112 participants (16.7%) with a history of gastrointestinal disorders, while the IBS group had 48 participants (25.9%) with history of gastrointestinal disorders. Showing a significant difference (*p* < 0.001) among the two groups. In addition, the distribution of the stress, anxiety, and depression case was significantly increased in the IBS group (*p* < 0.001).

### 3.2. Comparison of Nutrients Intake

Results of the comparison of nutrients intake between the two groups are shown in Table 2. Although no difference was observed in the energy intake between the two groups, the non-IBS group showed a fat intake of 76.9 ± 47.9 g/day, while the IBS group showed a fat intake of 86.6 ± 55.1 g/day, which was significantly higher (*p* = 0.014). In addition, the non-IBS group showed a total FODMAP intake of 12.6 ± 9.7 g/day, while the IBS group showed a total FODMAP intake of 13.9 ± 9.9 g/day (*p* = 0.030). The non-IBS group showed a fructan intake of 5.4 ± 4.3 g/day, while the IBS group showed a fructan intake of 6.1 ± 5.0 g/day (*p* = 0.013). The non-IBS group showed a mannitol intake of 0.4 ± 0.6 g/day, while the IBS group showed a mannitol intake of 0.5 ± 0.5 g/day (*p* = 0.025). The above findings demonstrated that the IBS group had a significantly higher intake of certain FODMAP nutrients. Additionally, there was no significant difference in nutrient intake according to IBS in male. However, in females, the fat intake of total energy was 22.9% in the non-IBS group and 24.8% in the IBS group (*p* = 0.001). A fructan intake was also significantly higher than the non-IBS group, 5.6 ± 4.6 g/day in the IBS group (*p* = 0.045).

### 3.3. Top 30 Food Items Causing Symptoms in IBS Patients

The food items causing gastrointestinal aggravation in IBS patients were analyzed and the results are shown in Table 3. These items were as follows: instant ramen (70.8%); Chinese-style noodles with vegetables and seafoods (68.7%); pizza (67.2%); black bean sauce noodles (66.3%); stir-fried rice cakes (tteokbokki) (64.0%); hamburger (62.9%); fried chicken (62.2%); pork belly (59.9%); takju (makgeolli) (59.5%); and sweet and sour pork cutlet (59.2%).

### 3.4. Food Group Causing IBS Symptoms

The level of contribution to the symptoms of IBS was analyzed by dividing the SQ-FFQ items based on the types of food causing the symptoms, and the results are shown in Table 4. A total of 186 participants (100%) showed symptoms when they consumed gluten; 164 (88.2%), when they consumed high fat foods and foods with high FODMAP levels; 140 (75.3%), when they consumed spicy food; 122 (65.6%), when they consumed dairy products; 115 (61.8%), when they consumed caffeine; and 101 (54.3%), when they consumed alcohol.

## 4. Discussion

This study performed in Korean adults in their 20 s to 40 s through a community-based web survey suggests that dietary intake and stress, depression, and anxiety differs between IBS patients and controls. The foods of high FODMAPs containing gluten were the most frequently reported food type affecting IBS symptoms, high fat food and high FODMAP food were the next most cited food, as reported by 88.2% of IBS patients.

In this study, Rome III was utilized to analyze the data acquired from 857 participants, and a prevalence rate of 21.7% was reported. A previous study that analyzed certain university students in groups reported a prevalence rate of 24.0–26.7% [5,6]. Although all previous studies selected Korean adults as participants and utilized Rome III for IBS diagnosis, this study reported a slightly lower prevalence rate, probably due to the differences in the study participants. In many studies, the proportion of participants in their 20 s and female was relatively higher; these results were similar to those of a Colombian study, which reported a value of 24% [32]. Age may have played a role in the lower prevalence rate obtained in the present study.

Previous studies on IBS have categorized dietary factors into dietary habits, dietary patterns, food groups, and nutrients. In terms of nutrients, high FODMAP intake had an influence on the incidence of symptoms. In this study, to establish dietary guidelines for Korean adults with IBS, dietary intake was examined, and nutrient intake was compared between patients with IBS and those without IBS. Although no significant difference was observed in energy intake between the IBS group and non-IBS group, the IBS group showed a significantly higher fat intake. This finding is consistent with the study of Tigchelaar and colleague that found that the IBS group showed a significantly higher fat intake than non-IBS persons in the Dutch population [33]. In a study conducted in the UK, female patients in the IBS group also showed a significantly higher intake of fats than the control [34]. Fat is associated with an increased gastric colon reflex [35] in patients with IBS. Fatty food slows the movement of gases in the small bowel and provokes greater rectal sensitivity, causing or worsening gastrointestinal symptoms. One interesting point is that there was a difference in fat intake in the non-IBS group and the IBS group according to sex. In a study investigating foods related to the onset of IBS symptoms in 197 patients with IBS, 52.3% of the respondents said fatty/fried food caused symptoms, and 142 persons of this study were women [36]. High fat intake releases cholecystokinin (CCK), which secretes bile acids that aid in fat digestion. As this CCK is affected by the menstrual cycle [37,38], it can change women’s fat absorption metabolism. Therefore, high fat intake should be considered to prevent symptoms of IBS in women.

In addition, FODMAP intake between the groups was compared. Results showed that the IBS group had a total FODMAP intake of 13.9 ± 9.9 g, which was significantly higher than the total FODMAP intake of 12.6 ± 9.7 g in the non-IBS group. Previous studies have reported that patients with IBS showed a total FODMAP intake of approximately 17–30 g [39,40,41,42]. Based on the results of this study, the overall FODMAP intake of Korean adults was relatively low. This finding was possibly due the high intake of dairy products in Western countries. A study analyzing the intake of FODMAPs in Spanish adults reported that adults aged 18–74 years showed a total FODMAP intake of 21.4 g and a lactose intake of 16.6 g, which accounted for 80% of the total FODMAP intake [15]. In this study, a lactose intake of total FODMAP intake was 4.8 g. On the contrary, the fructan intake in this study was relatively high (6.1 g), and it was attributed to the consumption of garlic, onion, green onion, etc., as seasonings have high fructan content.

The items that highly contributed to the occurrence of IBS symptoms were as follows: instant ramen (70.8%); Chinese-style noodles with vegetables and seafood (68.7%); pizza (67.2%); black bean sauce noodles (66.3%); stir-fried rice cakes (tteokbokki) (64.0%), hamburger (62.9%); and fried chicken (62.2%). They were mostly categorized as high FODMAPs with high fat contents. In a previous study, including 70 Brazilian female adults, the food items that caused gastrointestinal aggravations in IBS patients were fried food and food made from flour/wheat [43]. These results were partially similar to those reported in this study. Several studies reported that gluten can be a trigger for IBS patients. Additionally, wheat amylase trypsin inhibitors (ATIs) are implicated in the pathogenesis of IBS [44], as they elicit inflammatory and immune responses [27].

Our research has some limitations. We used a questionnaire that has never been validated for IBS patients, but this questionnaire used food items verified in the analysis of Koreans’ meal intake previously [19]. Additionally, the symptom patterns questionnaires were developed based on the Rome III standard after consulting a gastroenterologist. Another limitation was to analyze the food intake characteristics of IBS patients using a food frequency questionnaire. To evaluate the gastrointestinal symptoms for more specific foods or the pattern of symptoms, we need to perform further research using a 24 h recall method or food diary methods in the future. Additionally, this study was a cross-sectional study that included Korean adults in their 20 s to 40 s and participants were identified using only self-reported Rome III criteria for IBS diagnosis. Additionally, as this study only analyzed young adults in their 20 s to 40 s with high Western food intake, convenience food intake, and eat out frequencies, it is difficult to assume that these findings reflect the dietary intake status of all Korean patients with IBS. However, although domestic data on FODMAP intake are limited despite its close relation with dietary intake, this study is still significant in that it proposed the baseline data for establishing dietary guidelines by examining FODMAP intake in Korean patients with IBS and by identifying the food items that contributed to the occurrence of IBS symptoms.

Finally, the strength of this study is that, in Korean adults with IBS, which showed a lower intake of FODMAPs compared with the overseas patients with IBS, and the food items that highly contributed to the occurrence of symptoms had high fat and gluten content. The Korean population’s dietary culture, which involves the consumption of kimchi and vegetable side dishes, may increase the intake of FODMAPs. However, based on the results of this study, such food items were different from those causing the symptoms. Therefore, it seems unreasonable to directly apply the low FODMAP guidelines recommended in Western countries to Korean patients with IBS. However, since the intestinal absorption mechanism of FODMAPs could serve as a factor causing discomfort to patients with IBS, a follow-up study on specific food items that contribute to the high intake of FODMAPs must be conducted to establish dietary guidelines for patients with IBS.

## Figures and Tables

**Table 1 nutrients-13-01308-t001:** General characteristics between healthy controls and IBS patients ^a^.

Variables	Control		Cases with IBS		*p*-Value
	(*n* = 671, 78.3%)		(*n* = 186, 21.7%)		
Sex (male, *n* (%)) ^a^	350	(52.2)	78	(41.9)	0.016
Age (years) ^b^	30.8	±7.8	30.3	±7.3	0.357
BMI ^c^ (kg/m^2^)	23.1	±3.4	22.8	3.8	0.417
Obesity ^c^ (%)	177	(26.4)	45	(24.2)	0.549
Drinker (yes, *n* (%))	495	(73.8)	139	(74.7)	0.294
Smoker (yes, *n* (%))	158	(23.5)	41	(22.0)	0.210
Dietary supplement use (yes, *n* (%))	443	(66.0)	131	(70.8)	0.127
Medical history of related gastrointestinal diseases ^d^	112	(16.7)	48	(25.9)	<0.001
Food allergy (yes, *n* ((%))	52	(7.7)	22	(11.8)	0.103
Stress					<0.001
Low stress	220	(32.8)	34	(18.3)	
Middle stress	227	(33.8)	46	(24.7)	
High stress	224	(33.4)	106	(57.0)	
Anxiety					<0.001
Normal	493	(73.5)	98	(52.7)	
Mild anxiety	104	(15.5)	45	(24.2)	
Moderate anxiety	56	(8.3)	30	(16.1)	
Severe anxiety	18	(2.7)	13	(7.0)	
Depression					<0.001
Normal	126	(18.8)	24	(12.9)	
Mild depression	250	(37.1)	48	(25.8)	
Moderate depression	242	(35.9)	93	(50.0)	
Severe depression	55	(8.2)	21	(11.3)	
Total physically active (MET–min/wk)	3316.6	±5136.5	3143.6	±5774.9	0.693

IBS, irritable bowel syndrome; BMI, body mass index; MET, metabolic equivalent ^a^
*n* (%); ^b^ Mean ± standard deviation; ^c^ BMI ≥ 25 kg/m^2^; ^d^ medical history of related gastrointestinal diseases were gastrointestinal reflux disease, functional dysphasia, chronic constipation, functional diarrhea, stomach cancer and colon cancer.

**Table 2 nutrients-13-01308-t002:** Nutrients intake between healthy controls and IBS patients by sex ^a^.

Variables	Total					Male					Female				
	Control (*n* = 671)		Cases with IBS (*n* = 186)		*p*-Value ^a^	Control (*n* = 350)		Cases with IBS (*n* = 78)		*p*-Value ^a^	Control (*n* = 321)		Cases with IBS (*n* = 108)		*p*-Value ^a^
Energy (kcal) ^b^	3058.5	±1512.2	3194.5	±1586.6	0.235	3267.0	±1605.8	3366.1	±1444.1	0.244	2833.1	±1371.4	3070.5	±1677.7	0.337
CHO (g)	452.6	±218.1	452.6	±214.5	0.805	478.4	±228.0	482.7	±209.0	0.470	424.6	±203.5	430.8	±216.7	0.936
CHO of total energy (%)	61.6	±7.8	59.8	±7.6	0.001	61.6	±7.9	60.6	±7.7	0.112	61.6	±7.8	59.2	±7.6	0.002
Total sugars (g)	75.0	±57.5	78.5	±59.6	0.242	76.4	±61.8	75.5	±54.3	0.619	73.6	±52.6	80.7	±63.4	0.284
PRO (g)	117.9	±68.9	124.6	±71.7	0.124	126.2	±74.7	131.2	±64.4	0.123	109.0	±60.9	119.9	±76.5	0.301
PRO of total energy (%)	15.6	±2.7	15.9	±2.7	0.097	15.7	±15.9	15.9	±2.7	0.377	15.5	±2.6	15.9	±2.6	0.116
FAT (g)	76.9	±47.9	86.6	±55.1	0.014	81.5	±51.8	87.6	±46.4	0.063	72.1	±42.9	85.9	±60.9	0.057
FAT of total energy (%)	22.8	±5.9	24.3	±5.7	0.000	22.6	±5.8	23.5	±5.5	0.082	22.9	±5.9	24.8	±5.8	0.001
Total dietary fiber (g)	22.0	±14.6	23.1	±14.8	0.124	22.5	±15.5	22.9	±14.4	0.339	21.5	±13.6	23.3	±15.1	0.216
Water soluble (g)	6.2	±4.8	6.5	±5.0	0.209	6.2	5.1	6.2	±4.8	0.527	6.2	±4.5	6.6	±5.1	0.291
Insoluble (g)	15.2	±9.7	16.1	±9.8	0.088	15.7	±10.3	16.2	±9.6	0.240	14.7	±8.9	16.1	±9.9	0.314
FODMAP nutrients															
Total FODMAPs (g) ^c^	12.6	±9.7	13.9	±9.9	0.030	13.0	±9.9	14.2	±10.2	0.156	12.3	±9.5	13.6	±9.7	0.078
Fructan (g)	5.4	±4.3	6.1	±5.0	0.013	5.6	±3.9	6.8	±5.5	0.050	5.2	±4.7	5.6	±4.6	0.045
GOS (g)	0.1	±0.1	0.1	±0.1	0.310	0.1	±0.1	0.1	±0.1	0.231	0.1	±0.1	0.1	±0.1	0.627
Lactose (g)	4.7	±5.7	4.8	±5.4	0.488	4.7	±5.9	4.0	±4.3	0.518	4.7	±5.5	5.4	±6.0	0.156
Excess fructose (g)	0.9	±1.7	1.0	±2.1	0.546	0.9	±1.4	0.9	±1.6	0.496	1.0	±1.9	1.0	±2.3	0.757
Sorbitol (g)	0.9	±1.3	0.9	±1.5	0.999	0.8	±1.3	0.8	±1.2	0.652	0.9	±1.4	1.0	±1.6	0.914
Mannitol (g)	0.4	±0.6	0.5	±0.5	0.025	0.5	±0.5	0.5	±0.7	0.116	0.4	±0.7	0.4	±0.3	0.132

CHO, carbohydrates; PRO, proteins; FAT, fats; GOS, galacto-oligosaccharides.; ^a^ The data were analyzed using the Mann–Whitney test according to IBS classification; ^b^ mean ± SD; ^c^ total FODMAPs = fructan + GOS + lactose + excess fructose + solbitol + mannitol.

**Table 3 nutrients-13-01308-t003:** Top 30 food items causing symptoms in IBS patients.

No.	Food Items	Food Group	*n* (%)	
1	Instant ramen	High fat and high FODMAP, containing gluten	126	(70.8) *
2	Chinese-style noodles with vegetables and seafoods	High fat and high FODMAP, containing gluten	114	(68.7)
3	Pizza	High fat and high FODMAP, containing gluten	117	(67.2)
4	Black bean sauce noodles	High fat and high FODMAP, containing gluten	116	(66.3)
5	Stir-fried rice cakes (tteokbokki)	Spicy and high FODMAP, containing gluten	110	(64.0)
6	Hamburger	High fat and high FODMAP, containing gluten	105	(62.9)
7	Fried chicken	High fat and high FODMAP, containing gluten	112	(62.2)
8	Pork belly	High fat	109	(59.9)
9	Takju (makgeolli)	Alcohol	69	(59.5)
10	Sweet and sour pork/cutlet	High fat	106	(59.2)
11	Beer	Alcohol	87	(58.0)
12	Noodle	High FODMAP, containing gluten	100	(57.1)
13	Soju	Alcohol	76	(55.9)
14	Milk	High FODMAP and dairy	89	(54.3)
15	Corn flake with milk	High FODMAP and dairy	73	(50.7)
16	Korean cold noodles (naengmyeon)	High FODMAP, containing gluten	78	(46.7)
17	Spicy pork/bulgogi/galbi	High fat	84	(46.4)
18	Vegetable pancake	High fat and high FODMAP, containing gluten	77	(44.0)
19	French fries	High fat	74	(42.5)
20	Coffee creamer	Caffeine	36	(41.9)
21	Carbonated drink	High FODMAP and caffeine	73	(41.7)
22	Coffee	Caffeine	70	(41.7)
23	Sweet red bean bread	High FODMAP, containing gluten	62	(40.5)
24	Ice cream	High FODMAP and dairy	70	(40.2)
25	Sponge cake	High FODMAP, containing gluten	64	(39.8)
26	Boiled pork	High fat	67	(39.2)
27	Ham	High fat	68	(38.4)
28	Spicy sausage stew (budae jjigae)	High fat and spicy	62	(37.6)
29	Liquid yogurt	Dairy	61	(36.7)
30	Bibimbap	High FODMAP and spicy	63	(35.6)

* For each food item, 100% was based only on those who responded that they had eaten the food item.

**Table 4 nutrients-13-01308-t004:** Food groups causing IBS symptoms.

No.	Food Groups	*n* (%)	
1	High FODMAP, containing gluten	186	(100.0)
2	High fat	164	(88.2)
3	High FODMAP	164	(88.2)
4	Spicy	140	(75.3)
5	Dairy	122	(65.6)
6	Caffeine	115	(61.8)
7	Alcohol	101	(54.3)

## Data Availability

The database used and/or analyzed during the current study is not publicly available (to maintain privacy) but can be available from the corresponding author on reasonable request.

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
