# Peer review of "High-Fat Foods and FODMAPs Containing Gluten Foods Primarily Contribute to Symptoms of Irritable Bowel Syndrome in Korean Adults"

_nutrients, 2021, doi:10.3390/nu13041308_

Round 1

Reviewer 1 Report

The authors have addressed the main question analyzing the results by gender. The English has improved substantially but minor text editing is needed.

L49: Please, insert “Studies on food ingredients and irritable bowel…” instead of just “In food ingredients and irritable bowel…” otherwise, the sentence doesn’t make sense.

L56: Please, use capital W for “Western”.

In [line 52]: “In Western countries, grain bread, dairy products, and fruits with higher FODMAPs content are relatively higher than in Asian countries,…”

This sentence doesn’t appear in the revised paper as the authors state. It’s fine if they omit it. Otherwise, they can include it somewhere else.

L69: Please, use “One thousand…” instead of “On thousand…”.

L98: Please, insert “by adults in their 20s…” instead of just “by their 20s…”.

Author Response

30 March, 2021

Thank you for your comments about this manuscript. I modified the contents according to the reviewer's opinion. Modifications are shown in red mark on this manuscript.

Review 1

The authors have addressed the main question analyzing the results by gender. The English has improved substantially but minor text editing is needed.

L49: Please, insert “Studies on food ingredients and irritable bowel…” instead of just “In food ingredients and irritable bowel…” otherwise, the sentence doesn’t make sense.

  • We rewritten the sentence like this

: [line 49] Studies on food ingredients and irritable bowel syndrome are continuously actively conducted [15].  

L56: Please, use capital W for “Western”.

  • The word has been modified from lowercase to uppercase.

: [line 56] by Western countries.

In [line 52]: “In Western countries, grain bread, dairy products, and fruits with higher FODMAPs content are relatively higher than in Asian countries,…”

This sentence doesn’t appear in the revised paper as the authors state. It’s fine if they omit it. Otherwise, they can include it somewhere else.

  • As one of the suggested comments, We deleted the sentence.

L69: Please, use “One thousand…” instead of “On thousand…”.

  • The word has been modified the typo.

: [line 69] One thousand individuals~

L98: Please, insert “by adults in their 20s…” instead of just “by their 20s…”.

  • We rewritten the sentence like this

: [line 98] by adults in their 20s to 40s were added

Reviewer 2 Report

I am satisfied with the revision and Think that the authors have done a good work. I just have one suggestion. PLease rewrite the title. It is not good English and not very clear.

Author Response

I am satisfied with the revision and Think that the authors have done a good work. I just have one suggestion. Please rewrite the title. It is not good English and not very clear.

  • High-fat Foods and FODMAPs Containing Gluten Foods Primarily Contribute to Symptoms of Irritable Bowel Syndrome in Korean Adults

This manuscript is a resubmission of an earlier submission. The following is a list of the peer review reports and author responses from that submission.

Round 1

Reviewer 1 Report

Review Nutrients 210119

This is a very interesting article and an extremely important field of research. I think that the manuscript is very well written and structured. However, I have some concerns that I think deserve more attention.

  1. The symptoms questionnaire to assess GI symptoms, were those questionnaires validated? How did you assess these symptoms? BEPSI-K, HAD and IPAQ are very well described, but the symptoms questionnaire about GI symptoms deserve a better description, since the whole study is based on these symptoms.
  2. Results, page 4, You write on line 163-164 that “ The control group had 112 participants (16.7%) with a history of GI disorders, while the IBS group had 48 participants. Which type of GI disorders are you talking about? Please specify what this is. This is very important.
  3. Table 2. What is GOS? Please explain all abbreviations under the table.
  4. I also want to see the amount of ingested starch and sucrose, since these carbohydrates may be even more important for bowel health than lactose. Starch is ingested in higher amounts than other carbohydrates.
  5. Did you adjust the energy intake or food intake for sex? Since the gender distribution was unequal, you should adjust for sex in the calculations. Try regression analyses to be able to adjust for sex.

Author Response

Thank you very much for reviewing the manuscript carefully. We modified everything you pointed out. Modifications are shown in blue mark on the manuscript.

This is a very interesting article and an extremely important field of research. I think that the manuscript is very well written and structured. However, I have some concerns that I think deserve more attention.

1. The symptoms questionnaire to assess GI symptoms, were those questionnaires validated? How did you assess these symptoms? BEPSI-K, HAD and IPAQ are very well described, but the symptoms questionnaire about c, since the whole study is based on these symptoms.

- Line 110. We filled out a questionnaire for each food intake-related digestive symptom questionnaire using the same question as for the ROME III criteria.

2. Results, page 4, You write on line 163-164 that “The control group had 112 participants (16.7%) with a history of GI disorders, while the IBS group had 48 participants. Which type of GI disorders are you talking about? Please specify what this is. This is very important.

- Line 147. Subjects were asked to indicate the following gastrointestinal disorders on the questionnaire, such as gastrointestinal reflux disease, functional dysphasia, chronic constipation, functional diarrhea, stomach cancer, and colon cancer. Additional information about the GI disease-related questionnaire has been stated in the manuscript.

3. Table 2. What is GOS? Please explain all abbreviations under the table.

- Table 2, 3. The full name of GOS has been added in table 2 and 3.

4. I also want to see the amount of ingested starch and sucrose, since these carbohydrates may be even more important for bowel health than lactose. Starch is ingested in higher amounts than other carbohydrates.

- Table 2, 3. Thank the reviewer for what was pointed out regarding starch. However, there was no data on complex carbohydrates such as starch in the database used in this study. Only some monosaccharides, disaccharides, and dietary fiber data are available, so the data requested by the reviewer could not be added. Therefore, the analysis results for sucrose and maltose instead of starch are added to the table.

5. Did you adjust the energy intake or food intake for sex? Since the gender distribution was unequal, you should adjust for sex in the calculations. Try regression analyses to be able to adjust for sex.

- Table 3. Another reviewer gave the opinion that the table should be divided by gender. Therefore, we presented the table by dividing into man and women instead of adjusting the gender.

Reviewer 2 Report

The authors present a survey study to examine the main foods that contribute to IBS in Korean young adults. The study is well designed and organized, but some improvements should be made.

One main aspect is the English style, that needs improvement, and sometimes is hard to understand due to the structure of the sentences. The following corrections are proposed to the authors:

L36: please, delete "is one of the diseases with a high prevalence rate,"

L36/37: please delete "has a prevalence rate"

L41: Please insert "Based on previous studies," before "The British Dietetic ..."

L41-43: please delete "the results of previous studies were achieved after IBS patients followed evidence -based dietary guidelines, and such results contented that"

L49/50: Please, replace "such the mechanism of absorption of carbohydrates in the intestine" by "this"

L52: Please, replace "as" by "since"

L56: Please, consider "despite" instead of "although"

L56: Please, delete "an"

L61: Please, consider "the result of the" instead of "a result of "

L62-65: Please, rephrase last paragraph. I suggest "In this study, the dietary status of Korean adults in their 20s to 40s with IBS was examined, and their intake of FODMAPs and other types of foods that contribute to IBS symptoms was analyzed, to provide baseline data for establishing dietary guidelines for Korean patients with IBS."

L76: Please insert "the" before "questionnare" and delete "data" after "questionnare"

L77/78: Please move "Data acquired from 918 participants were used in this study," before "after excluding participants..." on L75

L93 Please, insert "a" before "web-based"

L99: Please, insert ", those" right before "with different"

L107: Please, use "When" instead of "In the case where"

L165 Please, consider using "Showing a significant difference (p<0.001) among the two groups" instead of "The two groups showed..."

L166/167: Please, replace "cases in the IBS group... (p<0.001)" for "stress, anxiety, and depression cases was significantly increased in the IBS group (p<0.001)

L206: Please, use "on" instead of "with"

L207: Please, replace "Gluten contained foods" for "Foods containing gluten"

L221: Please, use "the incidence of symptoms" instead of "symptom incidence"

L230: Please, insert "provoke" before "greater rectal"

L233/234: Please, delete spaces befre and after+- to keep uniformity in the manuscript

L248/249: Please delete "participants were divided into the IBS group and non-IBS group"

L250: Please, delete "than control"

L253: Please, replace "in inmplicated" for "are implicated"

L253: Please, insert "they " between "as elicit"

L256: Please, delete "that" before "the food"

L268: Please, replace "can inter" (doesn't make sense) for "can't infer"

Related to the results and analysis of the data:

L162: Please, correct "higher" to "lower"

IBS occurs more frequently in women than men, however the autors haven't analyzed or shown any data by separate. It would improve the quality of the paper if the they also include an analysis of the data for every gender, and test if their conclusions and statistical differences are mantained in both cases for the same foods, and they discuss accordingly, otherwise.

Author Response

Thank you very much for reviewing the manuscript carefully. We modified everything you pointed out. Modifications are shown in red mark on the manuscript.

The authors present a survey study to examine the main foods that contribute to IBS in Korean young adults. The study is well designed and organized, but some improvements should be made.

One main aspect is the English style, that needs improvement, and sometimes is hard to understand due to the structure of the sentences. The following corrections are proposed to the authors:.

Review comments

Modified line on manuscript

Modified sentence

L36: please, delete "is one of the diseases with a high prevalence rate,"

L36/37: please delete "has a prevalence rate"

L 40

IBS has a global prevalence rate of 9%-20%, and of 6.6%-26.7% in Korea.

L41: Please insert "Based on previous studies," before "The British Dietetic ..."

L41-43: please delete "the results of previous studies were achieved after IBS patients followed evidence-based dietary guidelines, and such results contented that

L 45

Based on previous studies, the British Dietetic Association reported that alcohol, caffeine, spicy food, fat, water in food, dietary habits, dairy products, dietary fiber, gluten, and high-sensitivity food may aggravate the symptoms

L49/50: Please, replace "such the mechanism of absorption of carbohydrates in the intestine" by "this"

L 52

Due to this, caution should be observed when consuming grain bread, dairy products, ~

L52: Please, replace "as" by "since"

L 54

since the intake of such foods is relatively higher in Western countries than in Asian countries,

L56: Please, consider "despite" instead of "although"

L56: Please, delete "an"

L 58

In Korea, although the prevalence rate exceeds the global average, IBS is only recognized as a symptom that causes discomfort; despite the occurrence of IBS is closely related to individuals’ dietary intake

L61: Please, consider "the result of the" instead of "a result of "

L 63

It can be assumed that the intake of FODMAPs (e.g., kimchi and vegetables) is the result of the consumption of different types of food in Western countries,

Please, rephrase last paragraph. I suggest "In this study, the dietary status of Korean adults in their 20s to 40s with IBS was examined, and their intake of FODMAPs and other types of foods that contribute to IBS symptoms was analyzed, to provide baseline data for establishing dietary guidelines for Korean patients with IBS."

L 64

In this study, the dietary status of Korean adults in their 20s to 40s with IBS was examined, and their intake of FODMAPs and other types of foods that contribute to IBS symptoms was analyzed, to provide baseline data for establishing dietary guidelines for Korean patients with IBS.

L76: Please insert "the" before "questionnare" and delete "data" after "questionnare"

L 79

the questionnaire such as IBS diagnosis,

L77/78: Please move "Data acquired from 918 participants were used in this study," before "after excluding participants..." on L75

L 78

Data acquired from 918 participants were used in the study after excluding participants who responded insincerely,

L93 Please, insert "a" before "web-based"

L 96

Participants completed a web-based semi-quantitative food frequency questionnaire (SQ-FFQ).

L99: Please, insert ", those" right before "with different"

L 102

among Korean type Chinese foods those with different cooking methods were classified into 2 items

L107: Please, use "When" instead of "In the case where"

L 111

When one of the food items caused the occurrence of symptoms,

L165 Please, consider using "Showing a significant difference (p<0.001) among the two groups" instead of "The two groups showed..."

L169

Showing a significant difference (p<0.001) among the two groups

L166/167: Please, replace "cases in the IBS group... (p<0.001)" for "stress, anxiety, and depression cases was significantly increased in the IBS group (p<0.001)

L 170

In addition, the distribution of the stress, anxiety, and depression case was significantly increased in the IBS group

L206: Please, use "on" instead of "with"

L 211

community based on web survey suggests that dietary intake

L207: Please, replace "Gluten contained foods" for "Foods containing gluten"

L 213

Foods containing gluten were the most frequently reported food type affecting IBS symptoms, ~

L221: Please, use "the incidence of symptoms" instead of "symptom incidence"

L 227

In terms of nutrients, high FODMAP intake had an influence on the incidence of symptoms.

L230: Please, insert "provoke" before "greater rectal"

L 236

Fatty food slows the movement of gases in the small bowel and provoke greater rectal sensitivity, ~

L233/234: Please, delete spaces befre and after+- to keep uniformity in the manuscript

L 246

Results showed that the IBS group had a total FODMAP intake of 13.9±9.9 g, which was significantly higher than the total FODMAP intake of 12.6±9.7 g ~

L248/249: Please delete "participants were divided into the IBS group and non-IBS group"

L 260

In a previous study including 70 Brazilian female adults.

L250: Please, delete "than control"

L 260

The food items that caused the gastrointestinal aggravations in IBS patients were fried food and food made from flour/wheat.

L253: Please, replace "in inmplicated" for "are implicated"

L 264

And wheat amylase trypsin inhibitors (ATIs) are implicated in the pathogenesis of IBS, ~

L253: Please, insert "they " between "as elicit"

L 264

[39], as they elicit inflammatory and immune responses

L256: Please, delete "that" before "the food"

L 267

, and the food items that highly contributed to the occurrence of symptoms had high fat and gluten content.

L268: Please, replace "can inter" (doesn't make sense) for "can't infer"

L 279

The sentence has been deleted because there is a similar phrase in the next paragraph.

Related to the results and analysis of the data:

L162: Please, correct "higher" to "lower"

L 166

which was significantly lower than the number of male participants in the control group (p=0.016)

IBS occurs more frequently in women than men, however the autors haven't analyzed or shown any data by separate. It would improve the quality of the paper if the they also include an analysis of the data for every gender, and test if their conclusions and statistical differences are mantained in both cases for the same foods, and they discuss accordingly, otherwise.

Table 3

Considering that IBS is a disease that occurs more often in female, according to the reviewer's opinion, the table was rewritten by dividing into male and female again.

The results of nutrient intake were presented separately between men and women in Table 3.

Round 2

Reviewer 1 Report

Dear authors, since you have not analyzed all the nutrient intake, only the content you are inteerested in, I can not accept this design. In Table 4, you write that the food groups high fat and gluten are included in these food items. But the food items contain so much more. You have not shown the content of starch and all other ingredients,e.g., fructan that are present in these items. In Table 3, it can be seen that fructan intake was higher in cases than in controls, and fructan is included in the same items as gluten. The conlucion is bias, since you had no open-minded aim and purpose of the study. 

Since there was no validated questionnaire regarding IBS sympotms, it is also unclear with the degree of symptoms. It is aslo starngge to included functional symptoms of diarrhea and constipaiton in the category previous GI disorders. Of course pateints with IBS have previosu dianosis of functional symptoms. Organic and fucntional previous disorders must be separated.

Author Response

Dear authors, since you have not analyzed all the nutrient intake, only the content you are interested in, I can not accept this design.

  • We thought deeply about this study through your review and made sincere corrections. I hope that we will consider once again whether our research is really appropriate. The revised contents are as follows. Modifications are shown in red mark on the manuscript.

In Table 4, you write that the food groups high fat and gluten are included in these food items. But the food items contain so much more. You have not shown the content of starch and all other ingredients,e.g., fructan that are present in these items.

In Table 3, it can be seen that fructan intake was higher in cases than in controls, and fructan is included in the same items as gluten. The conlucion is bias, since you had no open-minded aim and purpose of the study. 

  • We revised Table 2 by adding the analyzed nutrients, such as vitamins and minerals, according to the reviewer's opinion. As mentioned last time, there was no detailed food database on sugars in Korea, so other nutrients could not be added.
  • The classification of food types used in this study were classified based on IBS patients' nutrition guidelines reported by the British Dietetic Association. We categorized food types into alcohol, caffeine, spicy, fat, high FODMAPs, dairy, and gluten. In this study, evaluation of the intake tendency of high FODMAPs foods is also essential. Still, to reflect Korea's food type where flour intake is increasing, the food types were further classified into high FODMAPs, including gluten. A description of each food type classification has been added to the manuscripts. 
  • As a result of the analysis, this classification suggests that Korean IBS patients are related to IBS symptoms when they consume more FODMAPs containing gluten than to consume more FODMAPs, which are mainly emphasized in Western studies.
  • We wrote the limitations of our study additionally in the discussion part of the manuscript.

Since there was no validated questionnaire regarding IBS sympotms, it is also unclear with the degree of symptoms. It is aslo starngge to included functional symptoms of diarrhea and constipaiton in the category previous GI disorders. Of course patients with IBS have previosu dianosis of functional symptoms. Organic and fucntional previous disorders must be separated.

  • We have not been able to analyze the validity of the questionnaire assessed symptoms according to ingested foods. But the symptom-related questionnaire was prepared through consultation and review by some gastroenterologists, among the common symptoms of IBS. We also excluded subjects with inflammatory bowel disease and those who take drugs that can severely affect gastrointestinal motility (antibiotics, enema, anticonvulsant, anti-psychotic, and Parkinson’s treatment). Reflux esophagitis, functional indigestion, chronic constipation, and functional diarrhea were functional gastrointestinal diseases and could occur as symptoms of irritable bowel syndrome and were included in the subjects. Gastric cancer and colorectal cancer, which are organic gastrointestinal diseases, were included as subjects only when they were cured. We explained by adding the content to the manuscripts.
